**Data Availability Statement:** The data underlying the results presented in the study are available from (https://www.kaggle.com/martj42/international-football-results-from-1872-to-2017).

**Funding:** This work was supported by the EPSRC DTP studentship awarded to Matthew Penn for his

# Analysis of a double Poisson model for predicting football results in Euro 2020

**Matthew J. Penn** [1]*, **Christl A. Donnelly**[1,2]

**1** MPLS Division, Department of Statistics, University of Oxford, Oxford, United Kingdom, **2** MRC Centre for Global Infectious Disease Analysis, Department of Infectious Disease Epidemiology, School of Public Health, Faculty of Medicine, Imperial College London, London, United Kingdom

* matthew.penn@st-annes.ox.ac.uk

## Abstract

First developed in 1982, the double Poisson model, where goals scored by each team are assumed to be Poisson distributed with a mean depending on attacking and defensive strengths, remains a popular choice for predicting football scores, despite the multitude of newer methods that have been developed. This paper examines the pre-tournament predictions made using this model for the Euro 2020 football tournament. These predictions won the Royal Statistical Society's prediction competition, demonstrating that even this simple model can produce high-quality results. Moreover, the paper also presents a range of novel analytic results which exactly quantify the conditions for the existence and uniqueness of the solution to the equations for the model parameters. After deriving these results, it provides a novel examination of a potential problem with the model—the over-weighting of the results of weaker teams—and illustrates the effectiveness of ignoring results against the weakest opposition. It also compares the predictions with the actual results of Euro 2020, showing that they were extremely accurate in predicting the number of goals scored. Finally, it considers the choice of start date for the dataset, and illustrates that the choice made by the authors (which was to start the dataset just after the previous major international tournament) was close to optimal, at least in this case. The findings of this study give a better understanding of the mathematical behaviour of the double Poisson model and provide evidence for its effectiveness as a match prediction tool.

## Introduction

Predicting the results of an upcoming major football (soccer) tournament is often a matter of great public interest, with a wide variety of pundits [1] and professional companies [2] keen to voice their opinion. The factors considered in making these predictions are widely varied, taking into account many different aspects of the game of football such as individual players, tactics, past experience and form [1], but this paper aims to show that one can predict results with a high level of accuracy using a simple mathematical model based on past team performance.

The model will rely on the assumption that goals are scored according to a Poisson Process, an assumption seen in a wide variety of papers and first appearing in 1951 in [3]. Other models

DPhil in Statistics at the University of Oxford (https://www.ukri.org/councils/epsrc/career-and-skills-development/studentships/doctoral-training-partnerships/). The funders had no role in study design, data collection and analysis, decision to publish, or preparation of the manuscript.

**Competing interests:** The authors have declared that no competing interests exist.

**Abbreviations:** $c_B$, The number of goals conceded by Team; $E(X)$, The expected value of a random variable $X$; $f_A$, The number of goals scored by Team $A$ in the dataset $B$ in the dataset; $L(\boldsymbol{\theta})$, The likelihood of a dataset given the parameters $\boldsymbol{\theta}$; $\ell(\boldsymbol{\theta})$, The log likelihood of a dataset given the parameters $\boldsymbol{\theta}$; $M$, The set of matches in the dataset; $\mathbf{O}$, The vector of offensive strengths; $O_A$, The offensive strength of Team $A$; $P_{A,B}$, The number of matches played between teams $A$ and $B$ in the dataset; Poi $(\lambda)$, A Poisson variable with mean $\lambda$; $r_A$, The ranking of Team $A$ according to the double Poisson model; rA′, The ranking of Team $A$ according to the linear ranking model; $U(a, b)$, A random variable uniformly distributed on the interval $(a, b)$.; $\mathbf{V}$, The vector of defensive vulnerabilities; $V_B$, The defensive vulnerability of Team $B$; $\mu_{A,B}$, The expected number of goals scored by Team $A$ against Team $B$.

of goal-scoring have been used, such as in [4] (which edited the Poisson model to increase the probability of low-scoring draws), [5] (which uses Weibull rather than exponential inter-goal times) and [6] (which restricts its attention to predicting the goal difference in a game), while some models, such as the one given in [7] attempt to estimate result probabilities directly, without predicting the goals scored. However, the Poisson model has been shown to perform similarly well to a wide range of alternative models in studies such as [8, 9].

In order to calculate the means of these Poisson Processes, the method developed in [10] will be followed, which involves calculating an offensive and defensive strength for each team based on their previous results. Many papers, including [11] introduce an additional parameter to account for home advantage, but this will not be used in this paper due to the effects of the COVID-19 pandemic (during which a large number of matches in the dataset were played) —many papers, including [12], have shown that home advantage was significantly weakened, at least in domestic football, although [13] shows it still persisted to some extent. However, in non-pandemic times, it would be sensible to include such a parameter in the model.

The main novel contribution of this paper to the literature is a derivation of exact conditions under which the parameters in the model can be uniquely defined from maximising the likelihood. To the best of the authors' knowledge, these results have not appeared before. Another important novel contribution is the examination of the over-weighting of games containing defensively weak teams, which can skew the estimated offensive and defensive strengths in the model. Finally, by comparing the results of Euro 2020 to the predictions, it is possible to examine the weaknesses of the Poisson model and show where possible improvements could be made, while also illustrating that, despite these problems, it provided good predictions of the tournament.

This study is of course limited by the small number of models that it considers. Certainly, there is scope for a comparison of a much wider range of models, similar to the work done in [8, 9], which would help to show the extent to which the success of the double Poisson model in predicting Euro 2020 was due to the tournament having a small number of "suprising" results. However, the specific focus on the double Poisson model means that it can be examined in great detail, and both the analytic results derived, alongside the discussion of it's potential to over-weight games, provide useful information for those seeking to implement this model in the future.

A summary of the paper structure is shown in Fig 1. First, a brief derivation of the double Poisson model will be presented, alongside the final equations for the model parameters. Then, the conditions on existence and uniqueness of the solution will be stated, with the proofs left to the supplementary material. After this, the problem of high defensive vulnerability will be discussed, before the final predictions will be examined. This will be done by comparing them firstly to the results of Euro 2020. Then, the choice of start date for the dataset will be considered. Finally, the predictions will be compared to those made by a simpler linear model.

## Methods

### Model derivation

As previously stated, it will be assumed that each team scores goals according to independent Poisson Processes, so that

$$\text{Number of goals scored by Team A against Team B} \sim \text{Poi}(\mu_{A,B}) \tag{1}$$

where $\mu_{A,B}$ is the expected number of goals. Note that it is not necessary to assume that goals are scored at a constant rate throughout the match (i.e. that the Poisson Processes are homogeneous) because the final number of goals depends only on $\mu_{A,B}$—there is no requirement to

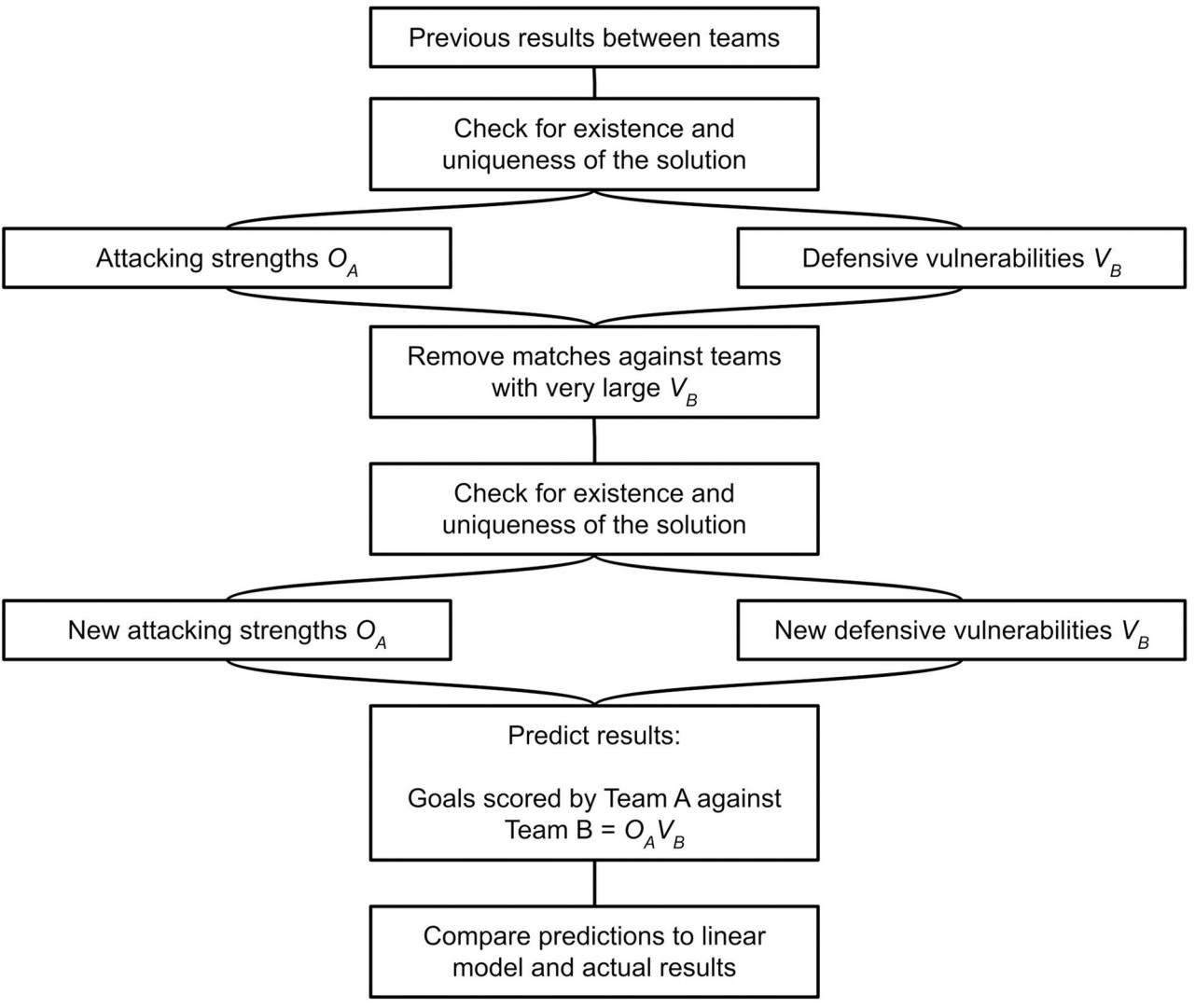

**Fig 1. The general sketch of this study.**

estimate when the goals will be scored. However, the use of Poisson Processes does require a "memoryless" assumption—that is, a goal being scored does not affect the number of goals scored in the rest of the game—which is in general not reflected in a football match, and is a potential source of error in the model.

It is also worth noting that, in this model, no distinction between home and away matches is made. That is, the expected number of goals that Team A scores against Team B will be equal to $\mu_{A,B}$ regardless of where the match is played. As discussed in the introduction, this is a simplifying assumption, but reduces the number of parameters and avoids over-fitting from the relatively small dataset used.

Indeed, due to the small number of relevant results between different teams (many of whom have not played each other at all) it is necessary to reduce the number of parameters in the model, rather than allowing each $\mu_{A,B}$ term to be independent. Thus, $\mu_{A,B}$ is assumed to be

of the form

$$\mu_{A,B} = O_A V_B \tag{2}$$

where $O_A$ denotes the attacking strength of Team $A$ and $V_B$ denotes the defensive vulnerability of Team $B$. The justification for this is found in the S1 File.

The $O_A$ and $V_B$ terms are not uniquely defined (one can see that multiplying the $O_A$'s by some positive constant $\lambda$ and dividing the the the $V_B$'s by the same constant will preserve the values of each $\mu_{A,B}$ term. However, their interpretation as offensive strengths and defensive vulnerabilities can be justified by noting

$$O_A = E(\text{goals scored by Team A against a team with defensive vulnerability 1}) \tag{3}$$

and

$$V_A = E(\text{goals conceded by Team A against a team with offensive strength 1}) \tag{4}$$

To estimate these parameters, maximum likelihood estimation will be used. Given a set of matches, $M$, represented by

$$M := \{(A, B, n) : \text{Match } n \text{ is between A (at home) and B (away)}\}, \tag{5}$$

define $x_h(n)$ to be the number of goals scored by the home team in match n and $x_a(n)$ to be the number of goals scored by the away team. Then, one can calculate the likelihood of the set of results (under the model) as

$$L(\boldsymbol{O}, \boldsymbol{V}) = \prod_{(A,B,n)\in M} \left( \frac{e^{-O_A V_B}(O_A V_B)^{x_h(n)}}{x_h(n)!} \right) \left( \frac{e^{-O_B D_A}(O_B D_A)^{x_a(n)}}{x_a(n)!} \right). \tag{6}$$

where, for example $\boldsymbol{O}$ represents the vector of offensive strengths. Note each term in the product is simply the Poisson probability of that number of goals being scored by the relevant team. This leads to a log likelihood of

$$\ell(\boldsymbol{O}, \boldsymbol{V}) = \sum_{(A,B,n)\in M} [-O_A V_B + x_h(n)\ln(O_A V_B) - O_B V_A + x_a(n)\ln(O_B V_A)] + \text{const..} \tag{7}$$

To find the optimal values of $O_A$ and $V_B$, it is necessary to solve the equations

$$\frac{\partial}{\partial O_A} = \sum_{\substack{A,B,n \in M \\ A \text{ fixed}}} \left[ -V_B + \frac{x_h n}{O_A} \right] + \sum_{\substack{B,A,n \in M \\ A \text{ fixed}}} \left[ -V_B + \frac{x_a n}{O_A} \right] = 0 \tag{8}$$

$$\frac{\partial}{\partial V_B} = \sum_{\substack{(A,B,n) \in M \\ B \text{ fixed}}} \left[ -O_A + \frac{x_h n}{V_B} \right] + \sum_{\substack{B,A,n \in M \\ B \text{ fixed}}} \left[ -O_A + \frac{x_a n}{V_B} \right] = 0. \tag{9}$$

These equations can be significantly simplified. Define

$$P_{A,B} := \text{Number of matches played between Team A and Team B}, \tag{10}$$

$$f_A := \text{Total goals scored by Team A}, \quad c_B := \text{Total goals conceded by Team B}. \tag{11}$$

Then, Eqs (8) and (9) become

$$-\sum_B P_{A,B} V_B + \frac{f_A}{O_A} = 0 \quad \text{and} \quad -\sum_A P_{A,B} O_A + \frac{c_B}{V_B} = 0. \tag{12}$$

These equations make sense—note that

$$-\sum_B P_{A,B} V_B + \frac{f_A}{O_A} = 0 \Rightarrow \sum_B P_{A,B} \mu_{A,B}(O_A, V_B) = f_A, \tag{13}$$

which results in the moment equation

$$\text{Expected Goals Scored By Team A} = \text{Total Goals Scored By Team A.} \tag{14}$$

Similarly, the other half of Eq (12) gives

$$\text{Expected Goals Conceded By Team B} = \text{Total Goals Conceded By Team B.} \tag{15}$$

showing that they are moment matching estimators.

## Existence and uniqueness of parameters

In general, there exist solutions to these equations, which give unique values of $\mu_{A,B}$. However, there are not unique values of the $O_A$ and $V_B$, as the model depends only on the product of these terms, and so redefining

$$O'_A = \rho O_A \quad \forall A \quad \text{and} \quad V'_B = \frac{1}{\rho} V_B \quad \forall B \tag{16}$$

means that the value of $\mu_{A,B}$ is unchanged for any $A$ and $B$. Thus, it is necessary to fix the value of one of the $O_A$ or $V_B$ terms in order to get a unique solution. In practice, this is not a problem when solving the equations—the solver will converge to a solution and effectively choose the scaling automatically—but it will be important to note this fact when calculating confidence intervals.

The exact conditions for existence and uniqueness of the solution are given in the following two theorems:

**Theorem 1** *Define the set of teams to be T. For any subset $S \subseteq T$, define $Q(S)$ to be the set of teams that have played at least one match against at least one of the teams in S. That is*

$$Q(S) = \{A \in T : \exists B \in S \quad \text{s.t.} \quad P_{A,B} > 0\}. \tag{17}$$

*Moreover, define*

$$G_{A,B} := \text{Total Number of Goals Scored by A against B,} \tag{18}$$

*and define R(S) to be the set of teams that at least one of the teams in S has scored against. That is,*

$$R(S) = \{B \in T : \exists A \in S \quad \text{s.t.} \quad G_{A,B} > 0\}. \tag{19}$$

*Then, there exists a finite global maximum of the log-likelihood $\ell(\boldsymbol{O}, \boldsymbol{V})$ defined in the S1 File, if and only if for any non-empty strict subset $S \subseteq T$,*

$$R[Q(S)] = S \Rightarrow R[Q(S) \cap Q(T/S)] = \emptyset. \tag{20}$$

**Theorem 2** *Define*

$$F := \{A \in T : f_A > 0\} \quad and \quad C := \{A \in T : c_B > 0\}. \tag{21}$$

*Suppose that F (and hence C) is non-empty. Then, the values of $\mu_{A,B} = O_A V_B$ are the same at each local maximum if and only if a finite maximum exists and for each non-empty set S,*

$$S \subset F \Rightarrow S \subset Q[Q(S) \cap C] \cap F \tag{22}$$

and for any $B \in T$

$$B \notin F \Rightarrow Q(B) \cap C \neq \emptyset \tag{23}$$

*and*

$$B \notin C \Rightarrow Q(B) \cap F \neq \emptyset. \tag{24}$$

The proofs of these theorems are long and technical and are given in the S2 File. However, combined, they show that there must be a unique solution for the parameters $\mu_{A,B}$ which must in turn occur at global maximum of $\ell$ (as there can only be one stationary point and $\ell$ must have a maximum). Alongside the proofs, algorithms are provided in S3 File. to efficiently check whether the conditions hold and thus providing a method to check whether a unique solution exists.

## Parameter estimation

**Data processing.** The dataset used to estimate $O_A$ and $V_B$ comprised of all results in international matches between European nations between July 16, 2018 (the day after the 2018 FIFA World Cup ended) and May 28, 2021, which were dowloaded from [14]. The start date was chosen as international teams tend to have significant transitions after major tournaments, particularly World Cups, and so was a compromise between the amount and the relevance of the data. Further investigation into this choice is found in the results section. The end date was chosen for administrative reasons (in order to submit the predictions in time for the competition), although it would have been preferable to include all pre-tournament games. Note that this dataset (and indeed all datasets) were found to have a unique solution for the $\mu_{A,B}$ from the algorithms derived in the previous section. The data was then processed using MATLAB R2020B and the code is available at [15].

**Initial estimation of $O_A$ and $V_B$.** Fig 2 shows the distribution of the values of $O_A$ and $V_B$. Note that these variables have been scaled in order that the maximum value of $V_B$ is equal to 1.

This figure suggests that the estimates are sensible—there is, as expected, a clear negative correlation between teams' attacking and defensive strengths. However, there is also a notable anomaly in the top-left corner—San Marino had an extremely low offensive strength of 0.09 and the maximum defensive vulnerability of 1 (much higher than the next highest, which was 0.79). This may perhaps seem unimportant—San Marino did concede a high number of goals and scored only once in the dataset—but, as will be shown in the next subsection, the large effect that this has on the other teams' strengths may be counter-productive.

**The problem of high defensive vulnerability.** If a team has a very high defensive vulnerability (in comparison to the other teams), the model parameters can become unreasonably skewed by the results between this team and the other teams. This means that it may be preferable to remove teams with high vulnerabilities from the model, as their games may not provide useful data on true team strengths. To highlight this, it is helpful to consider the limiting case of very defensively weak teams.

Suppose that the strengths are normalised such that there is some very weak team $M$ with the property that

$$V_M >> 1 \quad \text{and} \quad V_B \sim 1 \quad \forall B \neq M, \tag{25}$$

so that Team $M$ is significantly more defensively vulnerable than any other team. Then, consider the equation for the offensive strengths

$$\frac{f_A}{O_A} = \sum_B P_{A,B} V_B \sim P_{A,M} V_M \tag{26}$$

which is a good approximation provided

$$P_{A,M} V_M >> \sum_{B \neq M} P_{A,B} V_B. \tag{27}$$

Moreover, in this limiting case, one would expect most of the goals scored by each team to be scored in their games against Team $M$. That is,

$$f_A \sim G_{A,M} \tag{28}$$

and so,

$$O_A \sim \frac{G_{A,M}}{P_{A,M} V_M} \tag{29}$$

which means that the offensive strengths are to leading order determined solely by the results between teams $A$ and $M$. This is problematic as $G_{A,M}$ may in practice be roughly independent of $O_A$, as a strong offensive team is unlikely to perform better than a medium offensive team against team $M$—they will both score a large, fairly arbitrary number of goals.

This limiting case is somewhat unrealistic, but underlines a potential problem in the model —it gives high weight to games between very strong and very weak teams. This is a problem— for example, a result of 8–0 may lead to the strong team being rated significantly higher than if they had "only" won 5–0, even though both results are indicative of a similar level of domination. Indeed, in both cases, the intensity of the game is likely to drop significantly once the first few goals have been scored.

Because of this, all of the matches including San Marino were discarded from the model used for the competition, although both models will be analysed in this paper to test the effect of this decision. It would also have been possible to remove more teams—for example, Gibraltar's defensive vulnerability of 0.79 was approximately 0.07 higher than the next highest vulnerability—and further investigation may be useful in determining a threshold for removing teams.

A similar problem occurs if one team is significantly offensively stronger than all the other teams (as then the results against that team will determine the defensive vulnerabilities). However, as shown in Fig 2, there are no outliers with particularly high offensive strengths. Moreover, all of the teams with high offensive strengths were playing in Euro 2020, and so no teams were discarded based on their offensive strength.

**Final estimation of $O_A$ and $V_B$.** The values of $O_A$ and $V_B$ were thus re-estimated based on the results set with San Marino removed, and a comparison between the two sets of values is given in Figs 3–5.

These figures illustrate that removing San Marino most significantly changes the attacking strengths of the teams that played against San Marino. Indeed, all of the teams with attacking strengths that changed by at least 2% had played at least one match against San Marino. The

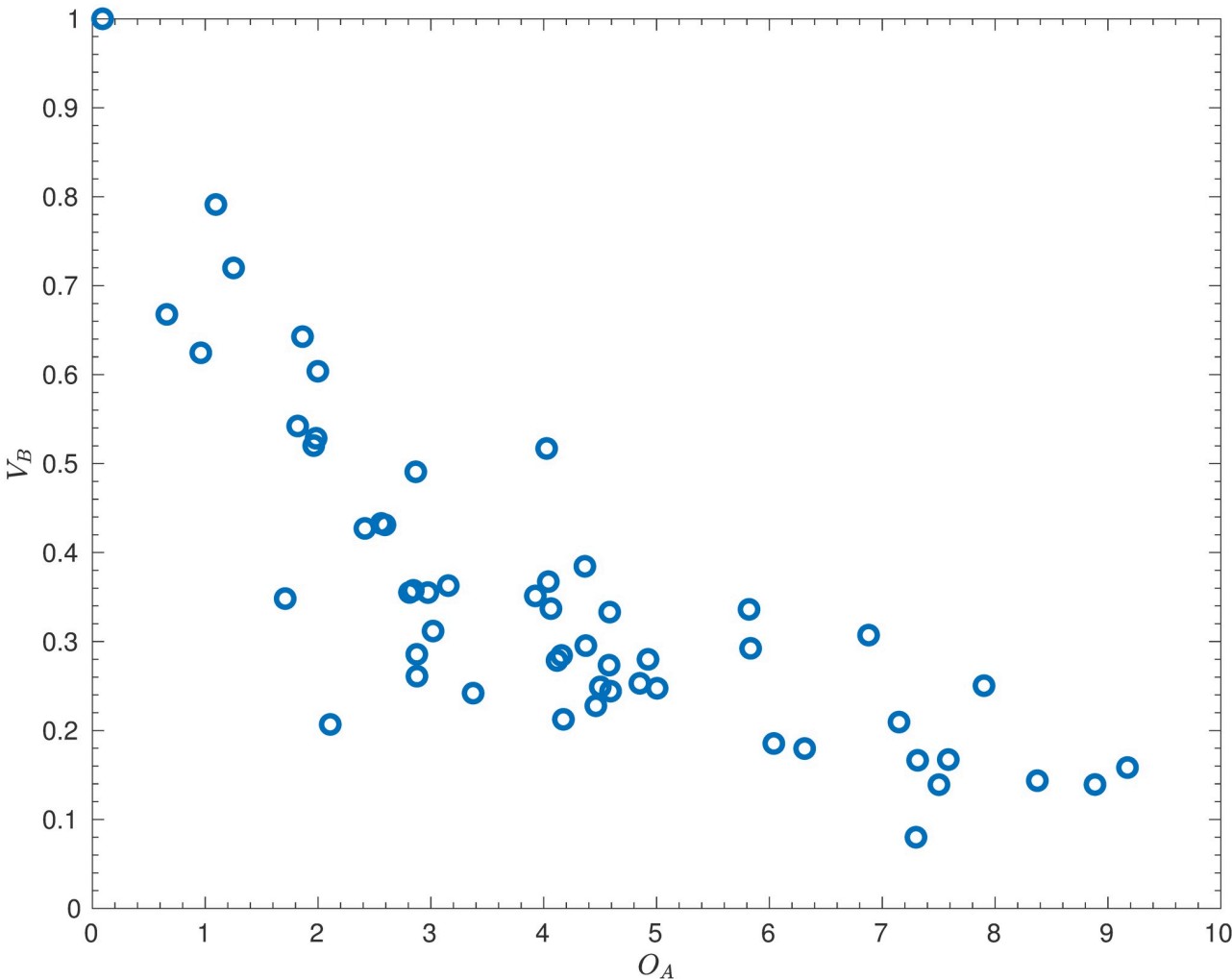

**Fig 2. Initial paramter values.** The values of the attacking strengths, $O_A$, and defensive vulnerabilities, $V_B$, from the full dataset of 55 European teams, using matches between July 16, 2018 and May 28, 2021. Note that the parameters have been normalised so that the maximum defensive vulnerability is equal to 1.

largest change was Cyprus, who scored 9 goals in two games against San Marino, 36% of their total goals across the 27 games they played in the original dataset. This illustrates the high weight of these games—their offensive strength falls dramatically once they are removed. Similar reductions in offensive strength were also caused to the offensive strengths of Moldova (who scored 27% of their goals against San Marino), Kazakhstan (25%), and other lower-ranked nations, albeit to a lesser extent. Moreover, the strengths of the strongest nations that had played against San Marino, Belgium and England, increased significantly as these countries had failed to reach the unrealistic expectation of averaging respectively 9.1 and 8.8 goals per game against San Marino (although they did achieve a very respectable 6.5 and 5.0 respectively).

The fact that removing San Marino had a much smaller impact on the estimated defensive vulnerabilities is positive—games against San Marino (who have such a low offensive strength that they are very unlikely to score against almost any opponent) provide very little information on the defensive vulnerabilities of their opponents. In fact, the differences at least 2%

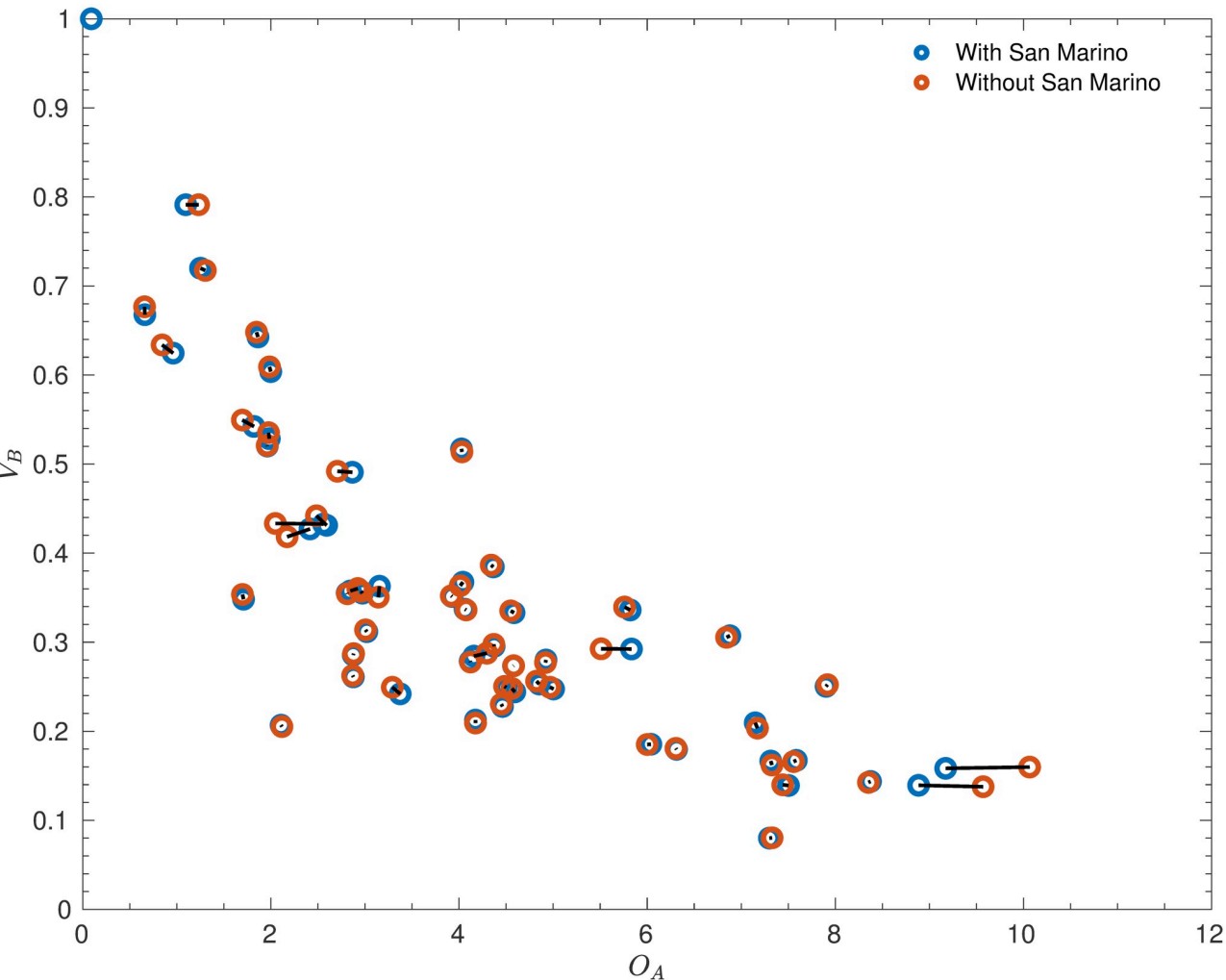

**Fig 3. Parameter changes after removing San Marino.** A comparison of the estimates of the offensive strengths, $O_A$, and defensive vulnerabilities, $V_B$, depending on whether San Marino's results are used. Note that the black lines join points corresponding to the same team, and that the variables have been normalised so that the defensive vulnerability of Gibraltar (0.79) is the same in both cases.

appear to be mostly secondary effects from the changes in offensive strengths—the largest change occured in a team (Iceland) that had not played San Marino!

The disadvantage of removing San Marino from the dataset is seen most clearly in the increase in offensive strength of Gibraltar, who managed to score only one goal in two games against San Marino, which, while lower than the expected 2.2 goals, was not particularly anomalous. However, it seems that the positive effects (particularly on the nations competing in Euro 2020) far outweigh the negative effects, and so San Marino were indeed removed in the final model. As will be shown in the Results section, this model performed marginally better because of this decision.

**Confidence intervals for $O_A$ and $V_B$.** Because $O_A$ and $V_B$ are maximum likelihood estimates (MLEs), it is possible to approximate confidence intervals using the information matrix. However, firstly, it is necessary to fix one of the variables as otherwise, there is still a degree of freedom in the model. This was done by fixing the defensive vulnerability of Gibraltar (the defensively weakest team left in the model) to be equal to 1. The confidence intervals are

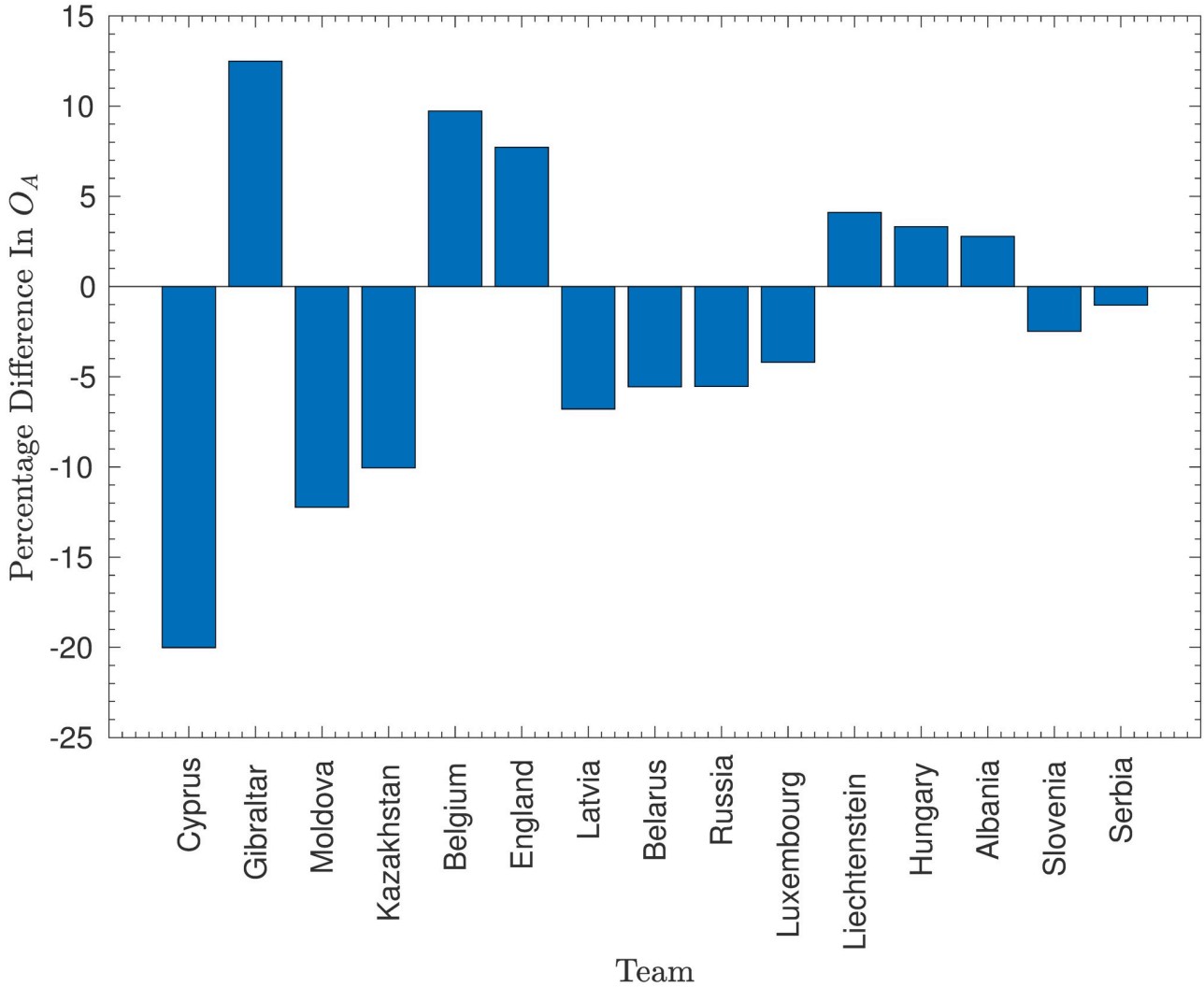

**Fig 4. Changes in offensive strength after removing San Marino.** The percentage change in the estimated offensive strength, $O_A$, for different teams when San Marino is removed from the model. Note that only values that changed by at least 2% have been included.

shown in Figs 6 and 7. Note that, in order to make the figures legible, only the teams playing in Euro 2020 have been included.

Figs 6 and 7 illustrate that there is a reasonable amount of uncertainty in the predicted strengths, with the confidence intervals for most pairs of teams overlapping. This is a disadvantage of using quite a small dataset, with only 1378 data points for the 108 model parameters. However there are still some significant differences between teams' strengths.

## Results

In order to test the efficacy of this model, it was used to create (pre-tournament) predictions for the delayed Euro 2020 competition, which ran from June 11 2021 to July 11 2021. The predictions were then entered into the RSS Euro 2020 Predictor Competition, [16], to assess them in comparison to other models.

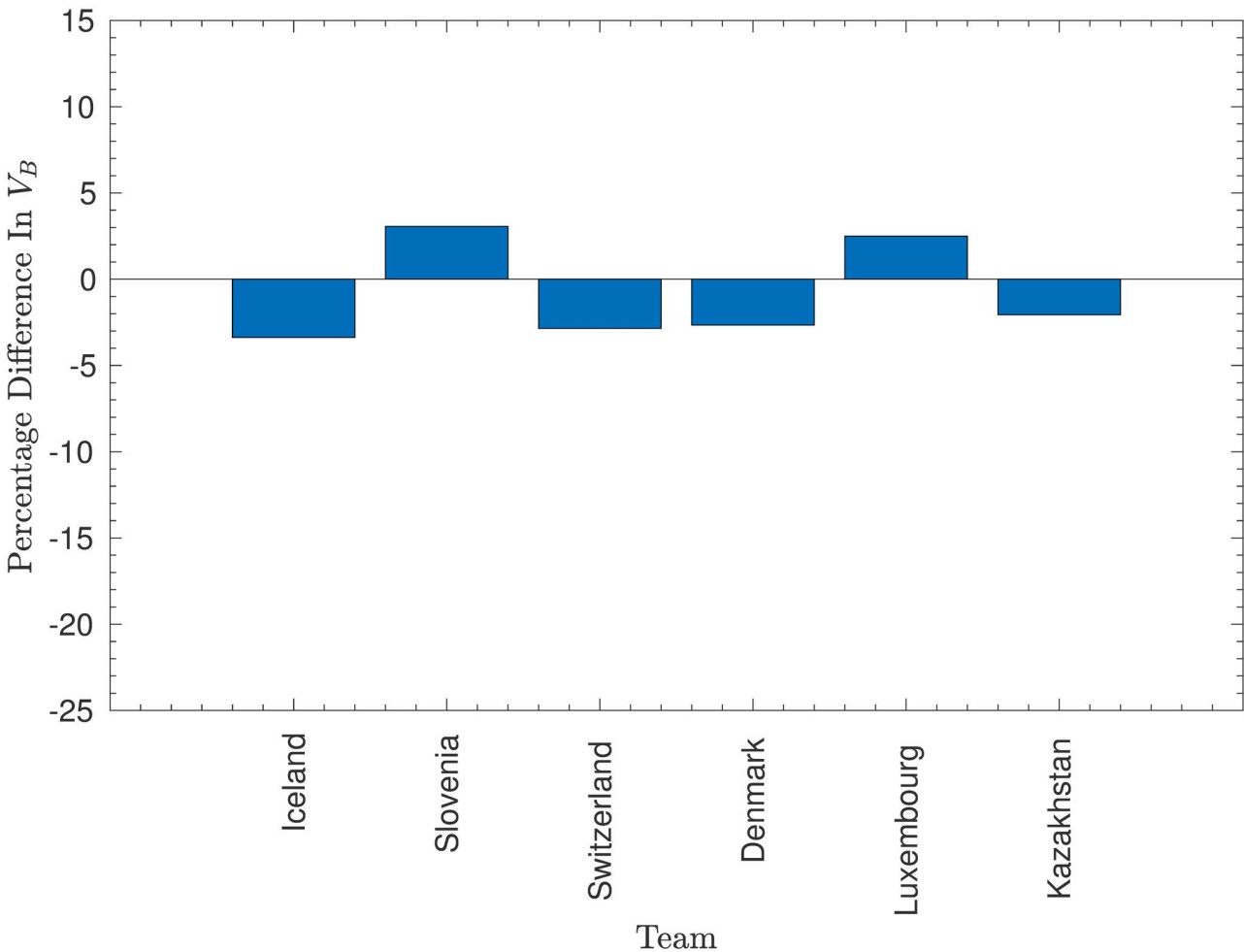

**Fig 5. Changes in defensive vulnerability after removing San Marino.** The percentage change in the estimated defensive vulnerability, $V_B$, for different teams when San Marino is removed from the model. Note that only values that changed by at least 2% have been included.

## Predicting goals

Figs 8 and 9 show a comparison between the predicted number of goals scored and conceded by each team and the actual values of these variables. These predictions were based on knowledge of which games actually took place in the tournament, alongside the games which went to extra time (and so were not pre-tournament predictions). Note that the prediction intervals for the predictions are calculated based on the assumption that the estimated parameters are correct (rather than incorporating the uncertainty in the parameters and that they were calculated so that the probability of the result lying below is at most 2.5% and the probability of the result lying above is at most 2.5%).

These predictions match the actual values very well, with 47 of the 48 actual values lying within the approximate 95% prediction intervals, and the majority of the values lying very close to the predictions (particularly in the predictions of goals scored). This is illustrated by the fact that the sum of the squared residuals across these 48 predictions is approximately 178, while the expected sum was 292. However, this large difference has a p-value of approximately 0.036, and so suggests that the model assumes too much variance in goals scored—something

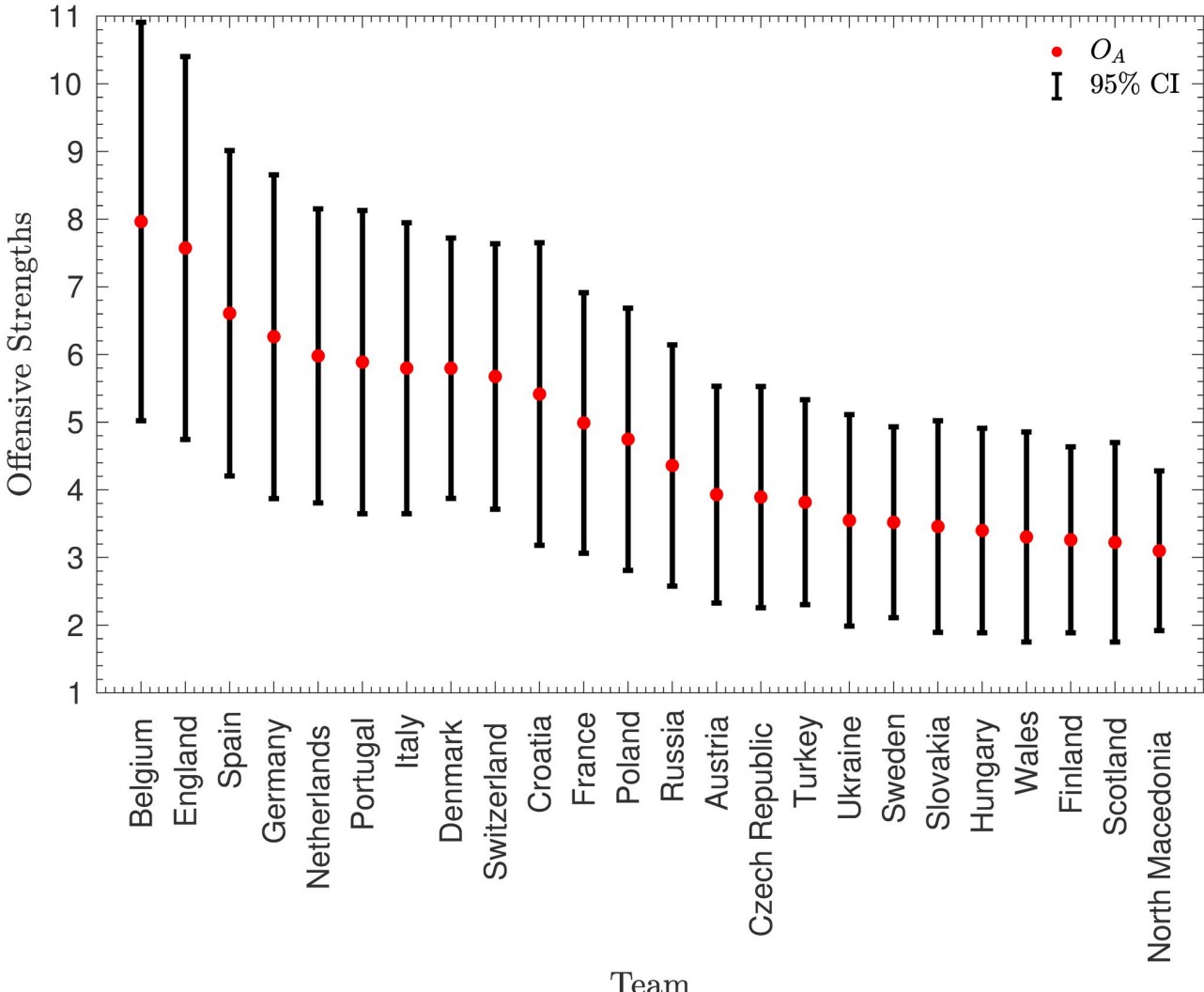

**Fig 6. Confidence intervals for the $O_A$ parameters.** Approximate 95% confidence intervals for the offensive strengths, $O_A$ of the teams playing in Euro 2020.

which is exacerbated by the fact that this variance does not include any parameter uncertainty. This is a common criticism of the Poisson model, with many authors such as [4] suggesting that the Poisson distribution over-estimates the probability of large numbers of goals, leading to a higher expected variance. However, it is encouraging that the assumption of Poisson variability does not appear to affect the means significantly, which appear to be good predictions.

## Predicting results

While goals scored and conceded provide a good way of assessing the effectiveness of the model, its purpose was to estimate the results achieved by each team, and these predictions will now be analysed.

One useful statistic is the log-likelihood of the set of results from Euro 2020 according to the model, the metric used in the RSS prediction competition [16]. This was -39.33, which

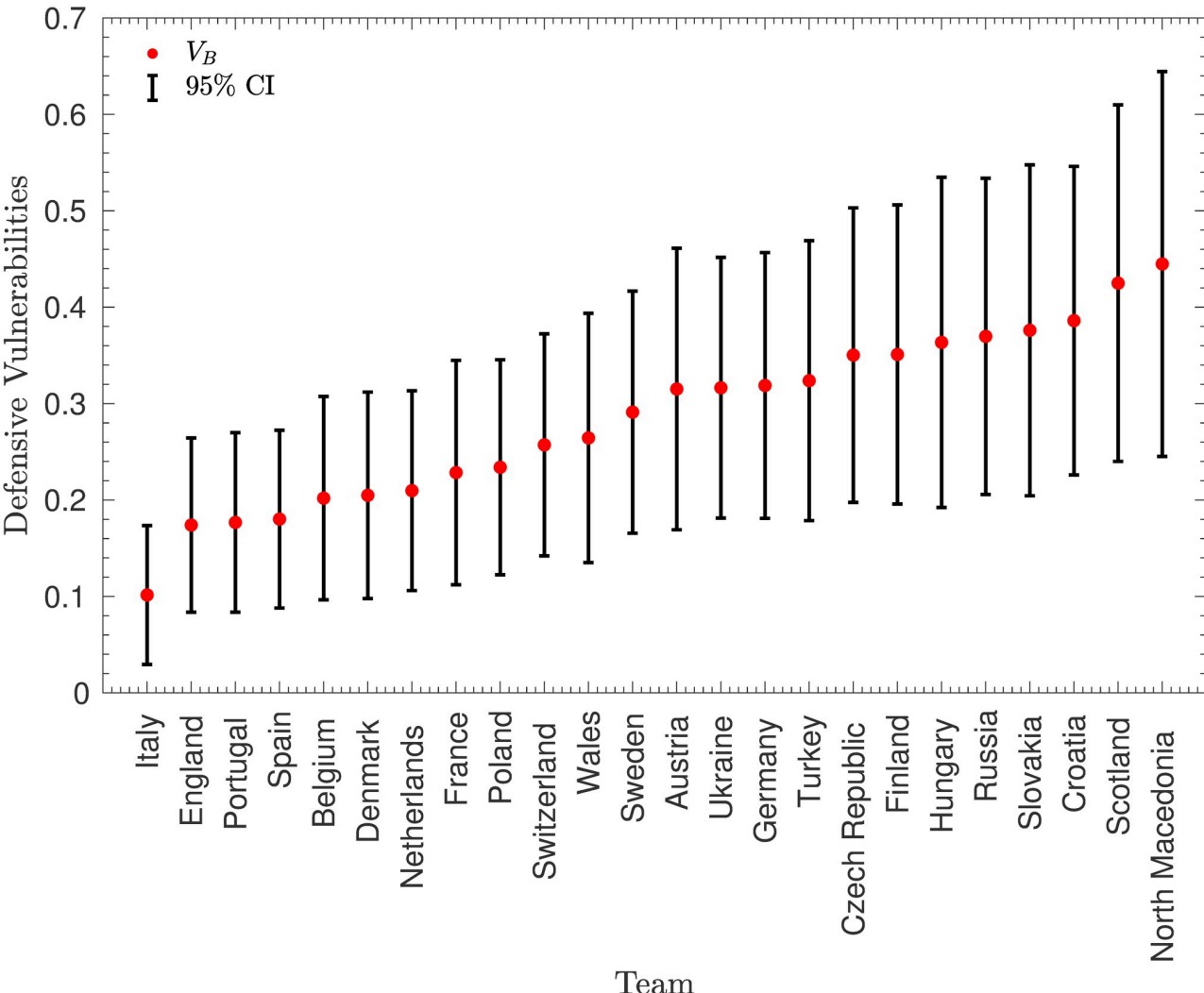

**Fig 7. Confidence intervals for the $V_B$ parameters.** Approximate 95% confidence intervals for the defensive vulnerabilities, $V_B$ of the teams playing in Euro 2020.

compares well with the expected value of the log-likelihood under the model (effectively the entropy of the tournament according to the model) which was -40.98. The probability of the log-likelihood being higher than -39.33 was approximately 0.34, suggesting that the problem of too much variance in the goals distribution does not occur as strongly when predicting individual results. Moreover, this model returned a higher log-likelihood than all of the other models in the RSS prediction competition, and indeed the log-likelihood of the model if San Marino had not been removed (which was -39.49) highlighting its ability to predict results accurately.

Another method of testing the predictions is to create a Quantile-Quantile (QQ) Plot. Given probabilities $p_n$, $q_n$ and $r_n$ for the three outcomes in match $n$, one can simulate the outcome of match $n$ using a uniform random variable $X_n$ and seeing which of the intervals $(0, p_n)$, $(p_n, p_n + q_n)$ and $(p_n + q_n, 1)$ it lies in. Conversely, given the outcome, one has a conditional

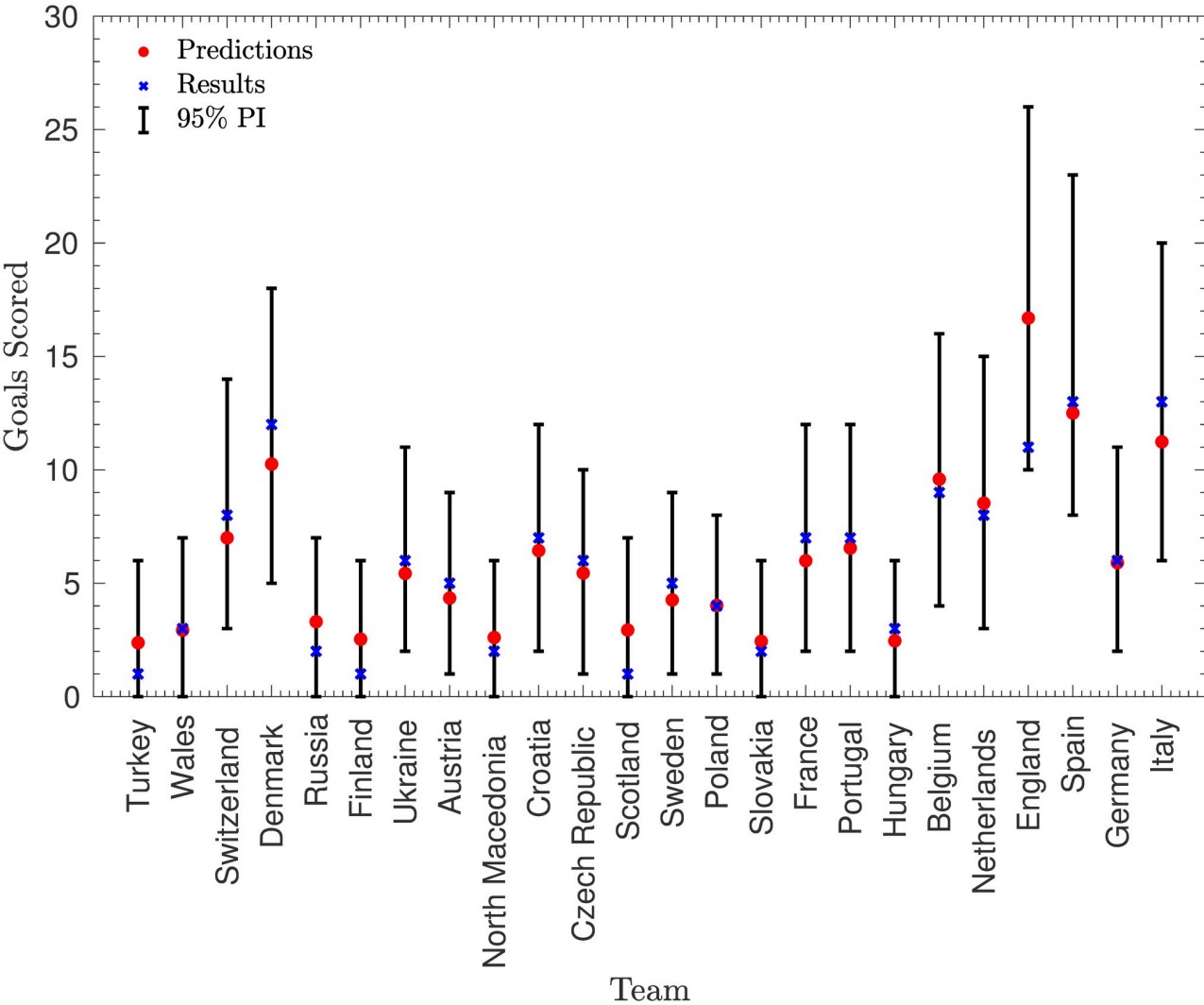

**Fig 8. Predicting goals scored by each team.** The number of goals scored by each team, alongside the predicted (mean) number of goals scored and an approximate 95% prediction interval.

distribution on $X_n$, which is that

$$
X_n \sim \begin{cases} U(0, p_n) & \text{if outcome corresponds to interval } (0, p_n) \\ U(p_n, p_n + q_n) & \text{if outcome corresponds to interval } (p_n, p_n + q_n) \\ U(p_n + q_n, 1) & \text{if outcome corresponds to interval } (p_n + q_n, 1) \end{cases} \tag{30}
$$

where $U(x, y)$ is a uniform random variable taking values in $x$ and $y$. Thus, one can re-simulate the variables $X_n$ using the outcomes of all of the matches, and create a QQ plot to test the fit. Moreover, by ensuring in each case that $p_n > q_n > r_n$, one can use this QQ plot to check for bias towards stronger or weaker teams.

Fig 10 shows a bias towards lower values of $X_n$, which in turn suggests a bias towards weaker teams in the model. That is, the stronger teams won more often than expected, which meant that the variable were sampled from a lower part of the interval [0, 1] than would be

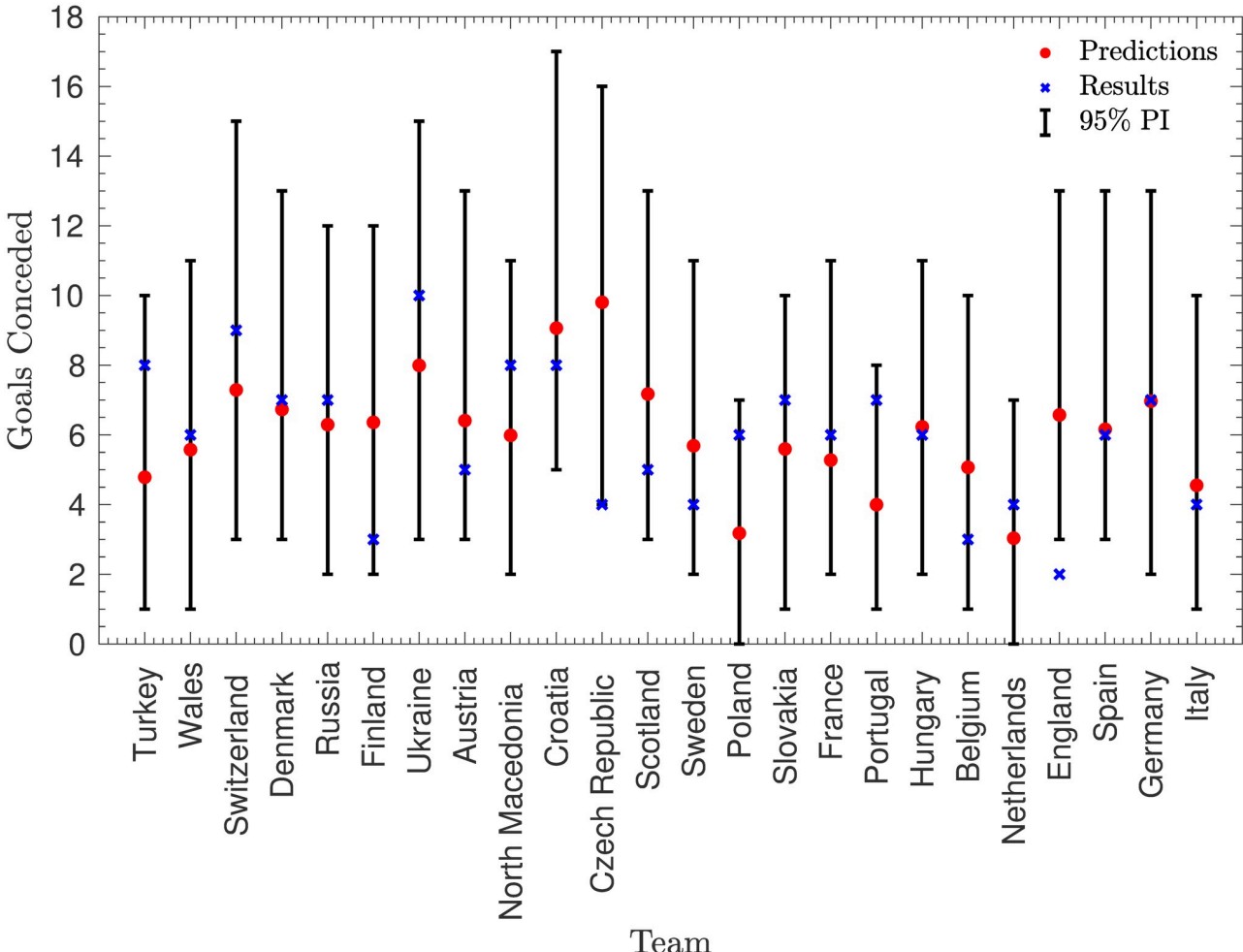

**Fig 9. Predicting goals conceded by each team.** The number of goals conceded by each team, alongside the predicted (mean) number of goals conceded and an approximate 95% prediction interval.

expected. This is in keeping with the fact that the log-likelihood was higher than expected (as the log-likelihood decreases faster when weaker teams perform better). However, recalling the very high p-value for the log-likelihood, further investigation is required to test whether this is simply random, or if the high variance in the Poisson goals distribution does cause the model to be biased towards weaker teams.

## Choosing the start date of the dataset

One of the most important decisions when using this method to model football results is to choose the time interval in which games are included in the dataset. The right endpoint of this interval is not difficult to choose—unless there are serious extenuating circumstances, more recent games should give more relevant data, and therefore the interval should include all games up to the date that the predictions are made. However, there are competing factors to consider when choosing the left endpoint. Increasing the number of games in the interval will decrease the effect of random noise, which can have a significant effect on the final predictions. However, the relevance of the data decreases as one moves further back in time (as the players

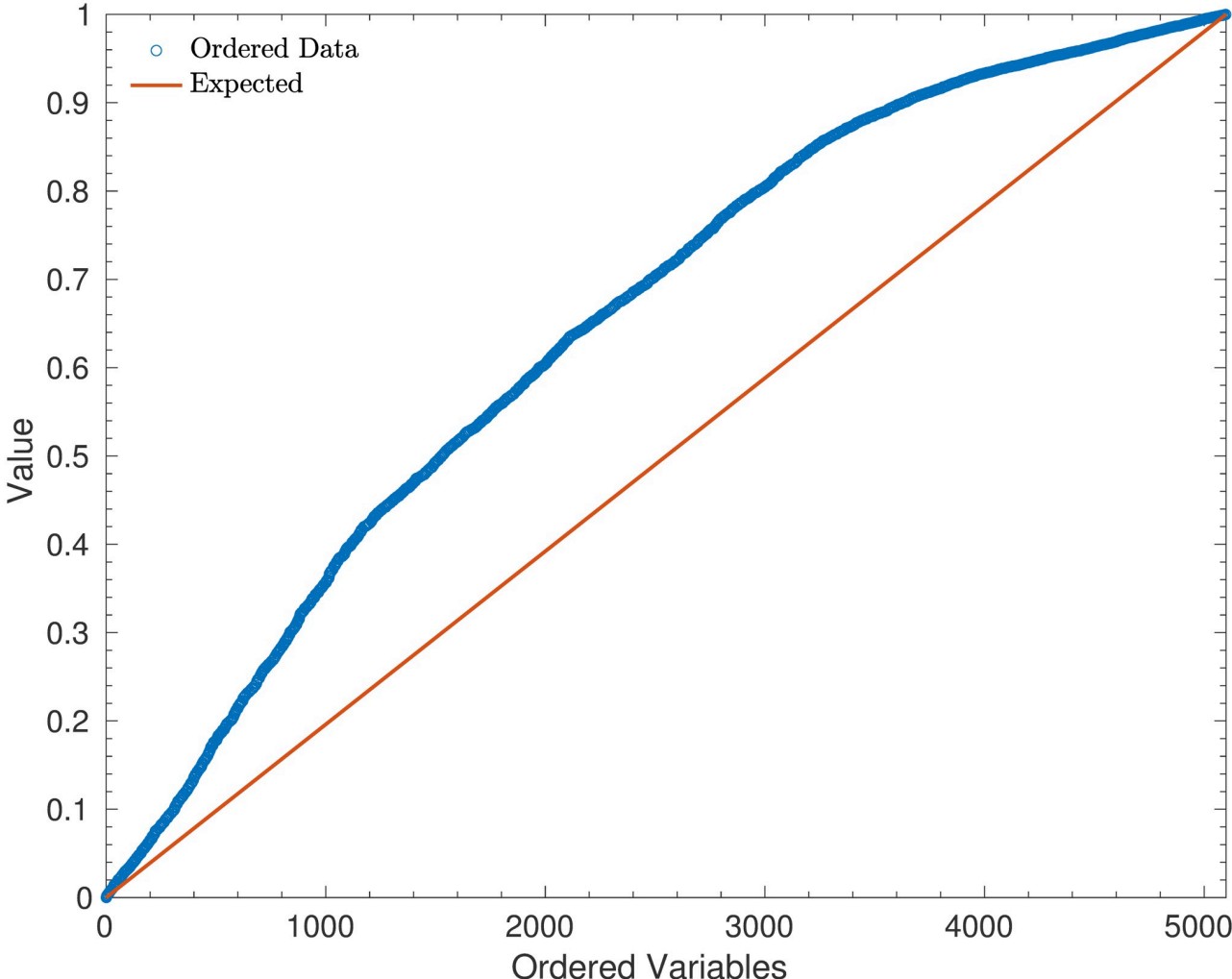

**Fig 10. A Quantile-Quantile (QQ) plot for the re-generated uniform random variables.** A QQ Plot for the uniform random variable simulations described in (30), conditional on the results of Euro 2020. Note that for each result, 100 uniform random variables were sampled in order to minimise variance in the plot.

and managers change) and so increasing the number of games may simply cause the final predictions to reflect past, rather than current strength.

Fig 11 illustrates how the predicted attacking strengths of the teams that competed in Euro 2020 changes based on the length of the dataset. It illustrates that the predictions are very sensitive to this length when the interval is small, but that this sensitivity decreases with interval length. It further shows that the final choice of the interval, which started on July 16, 2018, was in a region where this sensitivity was fairly low, suggesting that the interval was sufficiently long.

Fig 12 illustrates the score that would have been achieved depending on the start date of the dataset. This suggests that the choice of date made for the competition, July 16, 2018 was sensible, as the best score was achieved at October 10, 2018—very close to this date. Thus, it seems that the method of choosing the start date to be immediately after the previous major tournaments worked well in this case although perhaps one could afford to reduce the size of the dataset slightly.

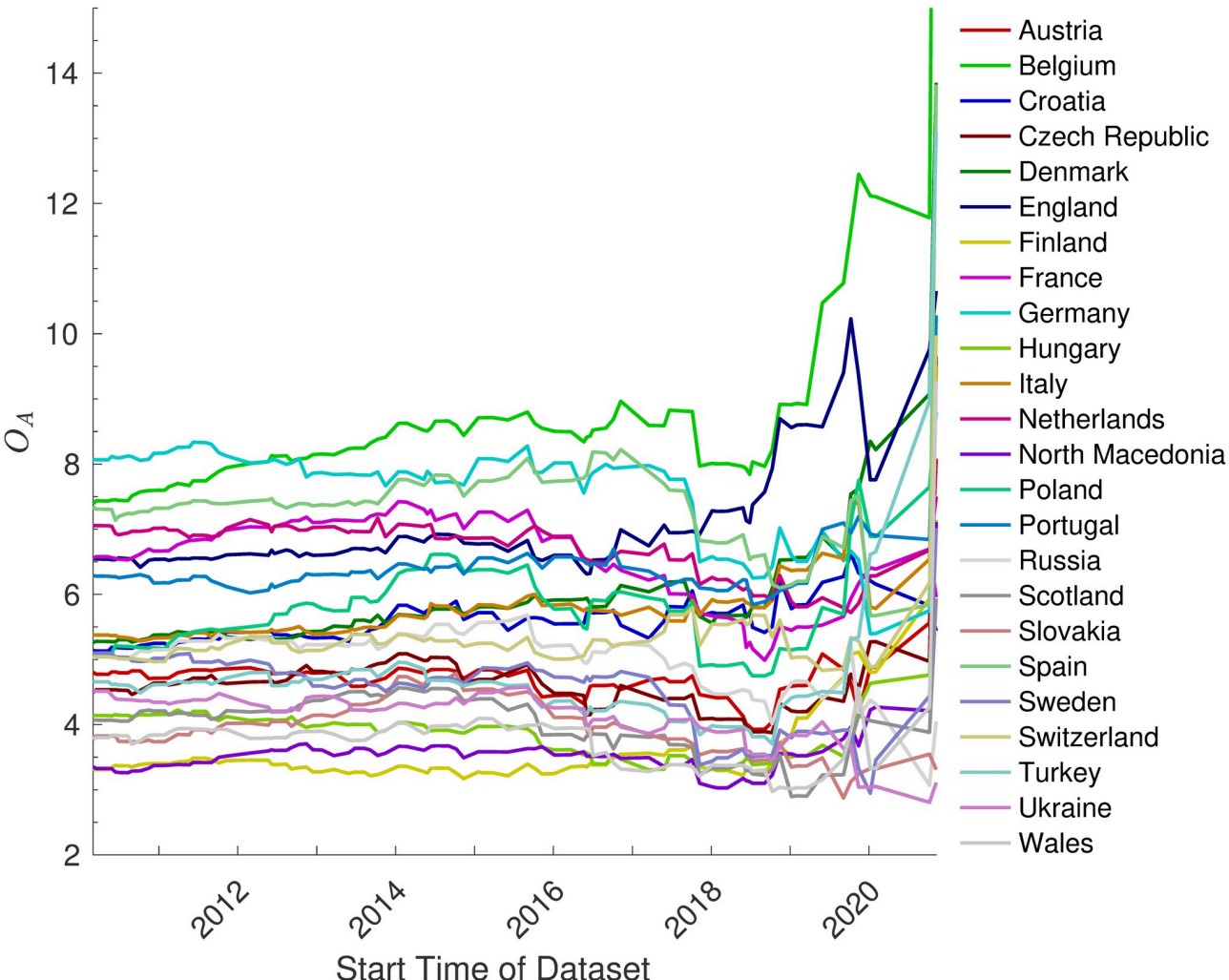

**Fig 11. The attacking strengths as a function of the start date of the dataset used.** This shows how the predicted attacking strengths, $O_A$ of the 24 teams that competed in Euro 2020 vary depending on the start date of the dataset of matches. Note that in all cases the end date was May 28, 2021—the date of submission to the competition.

It is worth noting that, were it not for the pandemic, there would have been fewer matches in the interval between the World Cup in 2018 and Euro 2020 (as the 2020–21 UEFA Nations League group stages were played before the Euro 2020 as a consequence of Euro 2020 being rearranged) as well as a shorter time interval (and therefore a lower level of player turnover). Further investigation would be required to see whether, in a "normal" international cycle, it would be beneficial to keep the interval at three years, rather than using the previous major tournament as the start date.

It is also worth noting that the log-likelihood appears to converge to approximately -43 as the length of the dataset increases to infinity. This is significantly higher than the log-likelihood of the "null" model (where each result is assumed to be equally likely), which is -49.9. This suggests that each country has some long-term advantage or disadvantage, which is plausible, as population and investment levels vary dramatically, and relatively stably, across the continent.

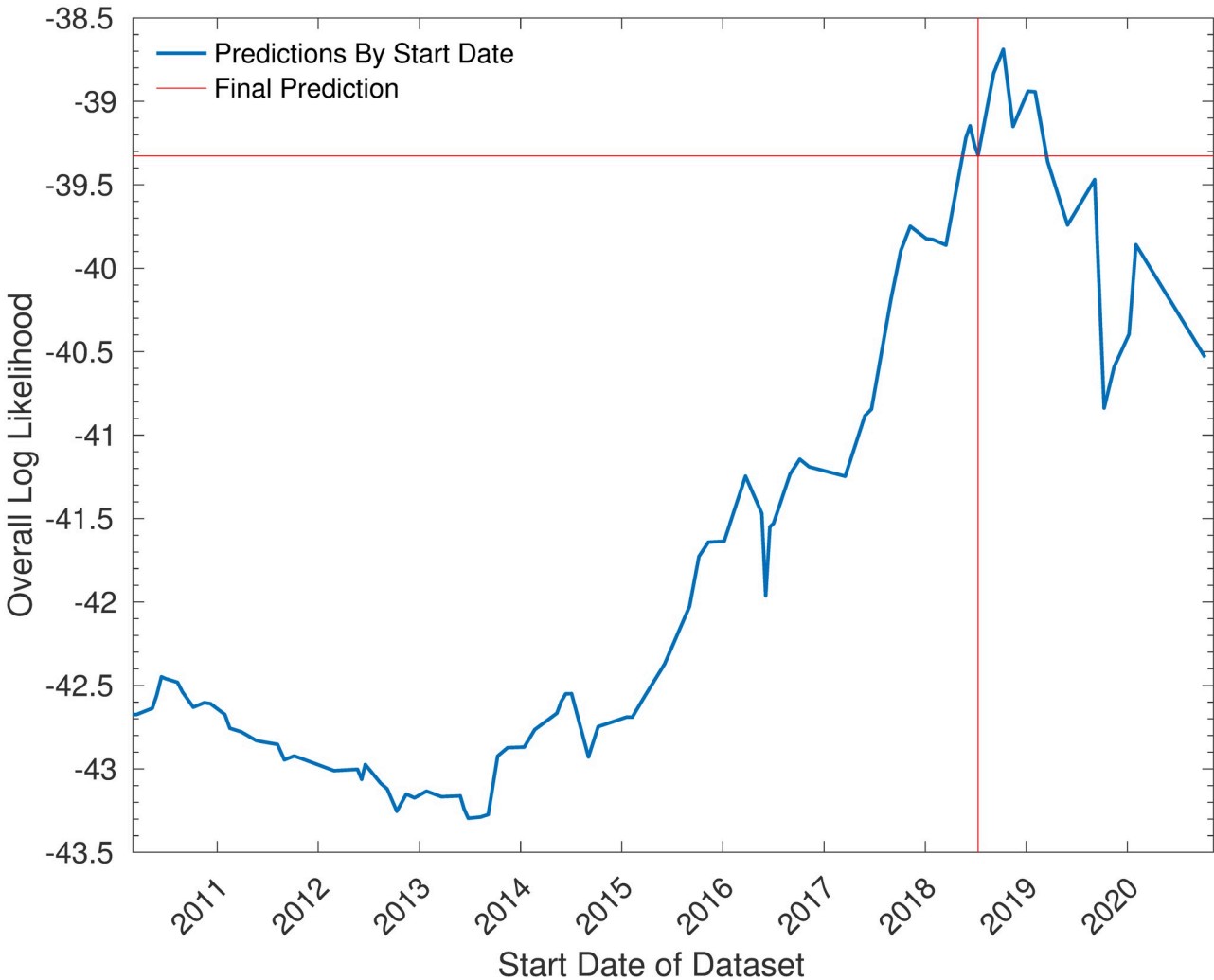

**Fig 12. The log-likelihood of Euro 2020 as a function of the start date of the dataset used.** This shows how the log-likelihood of Euro 2020 according to the model (the metric for the RSS competition) depends on the start date of the dataset of matches. Note that in all cases the end date was May 28, 2021—the date of submission to the competition, and each international break was considered as a single unit (and thus it was assumed that one would not set the start date in the middle of such a break).

## Comparison to a linear ranking model

An extension of the double Poisson model is that one can rank the teams based on the parameters $O_A$ and $V_B$. Indeed, note that

$$P(\text{Team A beats Team B}) > P(\text{Team B beats Team A}) \quad \Leftrightarrow \mu_{A,B} > \mu_{B,A} \quad (31)$$

$$\Leftrightarrow O_A V_B > O_B V_A \quad (32)$$

$$\Leftrightarrow \frac{O_A}{V_A} > \frac{O_B}{V_B}, \quad (33)$$

assuming that the defensive vulnerabilities are non-zero. Thus, one can define the ranking, $r_A$, of Team A to be

$$r_A = \frac{O_A}{V_A} \qquad (34)$$

One can test this model by comparing it to the linear ranking model derived in [17]. This model gives rankings, denoted here by $r_A'$ to teams, by assuming that when Team A plays against Team B, the number of goals scored by Team A subtracted by the number of goals scored by Team B (written as the goal difference, $d_{AB}$) is

$$d_{AB} = (r_A' - r_B') + \epsilon_{AB}, \qquad (35)$$

where $\epsilon_{AB}$ is some error term with mean 0.

Using least squares estimation then results in

$$r_A' = \frac{\sum_{B:(A,B,n)\in M} d_{AB}}{n_A}, \qquad (36)$$

where $n_A$ is the number of games played by Team A. This can be interpreted as

$$r_A' = \text{Average goal difference for Team A.} \qquad (37)$$

Fig 13 shows a comparison of the rankings for the 24 teams that took part in Euro 2020, after appropriate normalisation has taken place (noting that the Poisson rankings $r_A$ can be multiplied by any constant, and the linear rankings can be translated by any constant). The two rankings are relatively similar—there was an overall correlation of 0.92 between them—which suggests that the linear model retains most of the important information.

However, there are some notable exceptions which arise from the fact that the linear model does not take into account the difficulty of a team's fixture list. In particular, North Macedonia (the only team in this list that played in the bottom division of the UEFA Nations league, and therefore had weaker opponents more often) have a significantly inflated rank. This is exemplified by the fact that the correlation coefficient drops to 0.84 when the rankings of all modelled teams are compared—this list includes more teams whose ranking is increased like North Macedonia's. Certainly, the linear model relies on each team's fixtures being of comparable difficulty.

To compare the two models, one can use the goal difference in each match and compare it to the predictions. That is, if the actual goal difference for a match between Team $A$ and Team $B$ is $d_{AB}$, then

$$\text{Error for double Poisson model} = |d_{AB} - (O_A V_B - O_B V_A)|^2 \qquad (38)$$

and

$$\text{Error for linear model} = |d_{AB} - (r_A' - r_B')|^2. \qquad (39)$$

The results of Euro 2020 give a mean squared error of 2.05 for the Poisson model, lower than the mean squared error of 2.21 for the linear model. As the mean squared difference between the predictions is only 0.19, this suggests that, while the models give broadly similar results, the extra information used in the Poisson model does help it to be more accurate. However, more testing would be needed to determine this to a higher level of significance.

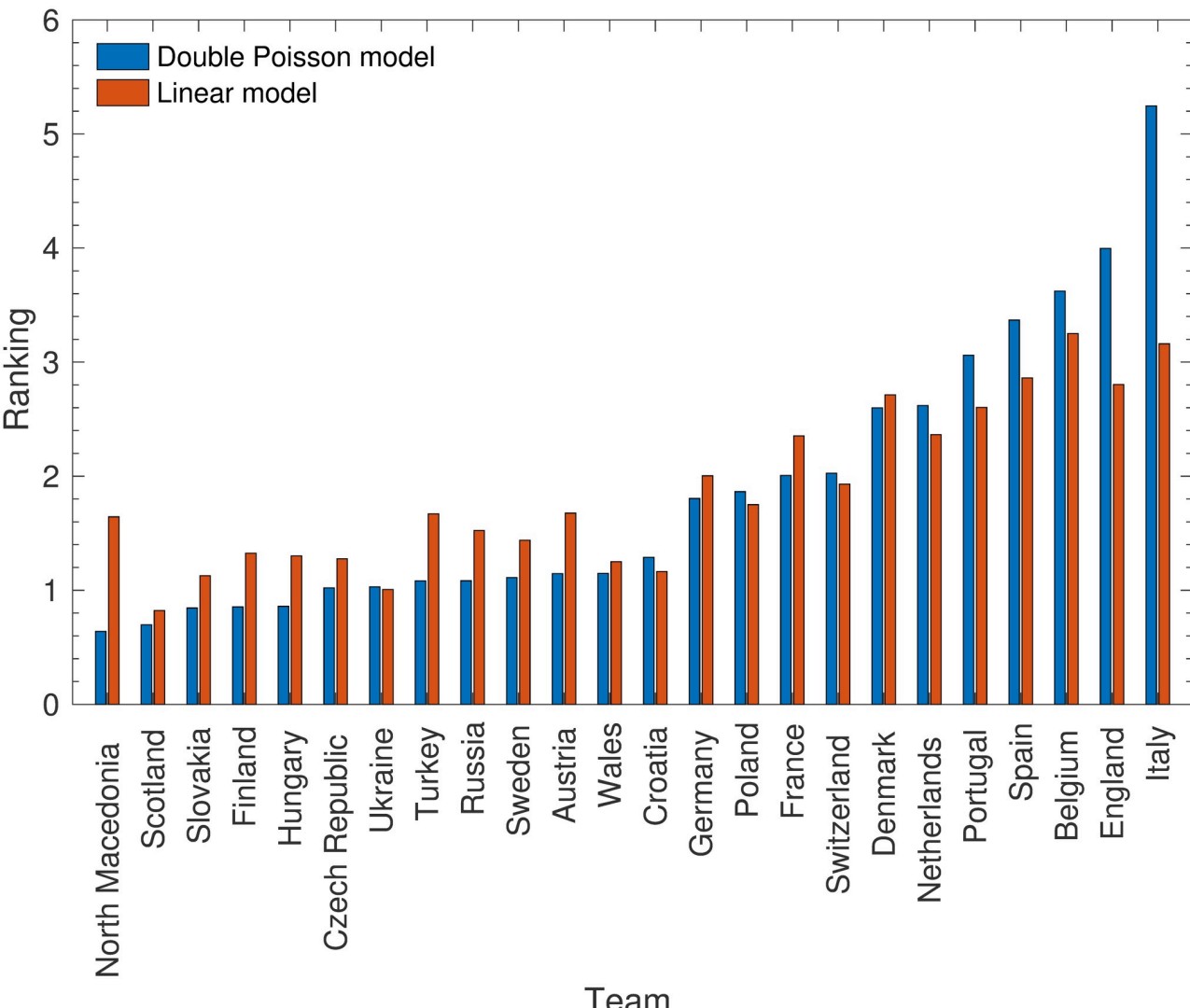

**Fig 13. A comparison of team rankings from the double Poisson model and those from the linear model.** This shows the rankings that are derived from each model for the 24 teams in Euro 2020. Note that the rankings have been normalised to ensure that they have the same sum, and to allow for easy comparison. This means that the Poisson rankings do not match the parameters considered in previous figures.

## Discussion

While derived from simple principles, the model presented in this paper gave predictions of Euro 2020 with a good level of accuracy. However, it appears that there are still improvements that could be made, as the data suggest that the model assumes too high a level of variance in the goals scored by each team, and in the overall results.

Alongside the results of the predictions, a wide range of analytic results were provided. An advantage of the Poisson Process model is that the conditions for existence and uniqueness of the parameter solution can be exactly formulated. This is especially useful when attempting to apply the model to small or weakly-connected datasets, as the algorithms given in this paper provide a simple way of determining whether sensible predictions can be made. This has potential applications in the scheduling of tournaments where there is insufficient time for a round-robin to be played, allowing the organisers to estimate the probabilities of there being

sufficient data for a ranking of the competitors to be formed in this way. The comparison in the previous section suggests this ranking would be more accurate than one formed from a linear model, increasing the utility of these results.

One way in which the model could be improved would be to use a different distribution for the number of goals scored. One simple change would be to set a maximum number of goals that a team could score in a game, which would help to eliminate the variance caused by the tails of the Poisson model. Alternatively, one could use a different underlying model altogether, for example, the Weibull model suggested in [5].

Furthermore, it would be beneficial to investigate whether weighting games in the dataset differently could help to improve the accuracy of the predictions. For example, Czech Republic performed far better than the model expected them to, but had much better results in the second half of the dataset compared with the first, suggesting that there could be a benefit to weighting more recent games more highly. Furthermore, the dataset contains a large number of friendly matches which are often played between second-string teams and may not provide much useful information about the result between those teams in a competitive match.

A further area of development would be to attempt to model the effect of home advantage, which was in [13] shown to be important even during the COVID-19 pandemic, when fans were unable to attend matches. Home advantage has been modelled in numerous papers, including [4], and may have been especially useful due to the unique nature of Euro 2020, where many teams played a couple of matches at home stadia.

However, this idiosyncrasy of Euro 2020 also made modelling easier, as it meant that all teams took part in the qualification process (unlike in "normal" tournaments where the host country qualifies automatically). To model the strengths of the host country accurately would be difficult, as there would be far less relevant data, and would possibly require the dataset to be taken over a longer interval, at least to estimate the strengths of this team. Further investigation would be required to quantify the effect of this on the accuracy of the overall predictions.

## Summary and conclusions

The results of this paper are summarised below

- The double Poisson model predicted the results of Euro 2020 with a high degree of accuracy.

- When using the double Poisson model, it may be helpful to remove results against teams with a very high defensive vulnerability, as these results can skew the dataset.

- The choice of start date for the model, which was the end of the previous major tournament, was close to optimal, and this could be a good guideline for future predictions.

- While the simple linear model also gave good predictions, it appears that the extra information used by the double Poisson model helps it to be more accurate in general.

- The double Poisson model may assume too much variance in the results, meaning that another distribution for goals scored could be more effective.

- Changes to the model, such as weighting the games differently depending on whether they are recent or important matches, could help to improve the accuracy of predictions

## Supporting information

**S1 File. Model derivation.**
(ZIP)

**S2 File. Proof of Theorems 1 and 2.**
(ZIP)

**S3 File. Algorithms for checking existence and uniqueness.** This presents the algorithms for checking existence and uniqueness and proves that they give the correct results.
(ZIP)

## Acknowledgments

The authors would like to thank the organisers of the prediction competition for providing the motivation for this model's development and the opportunity to compare it against others. They would also like to thank Cameron Bell for his invaluable proof-reading work.

## Author Contributions

**Conceptualization:** Matthew J. Penn.

**Formal analysis:** Matthew J. Penn.

**Investigation:** Matthew J. Penn.

**Methodology:** Matthew J. Penn.

**Software:** Matthew J. Penn.

**Supervision:** Christl A. Donnelly.

**Visualization:** Matthew J. Penn.

**Writing – review & editing:** Christl A. Donnelly.

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
