## [Editor Report · Decision Letter 0]

26 Jan 2022

PONE-D-22-01098Analysis of a double Poisson model for predicting football results in Euro 2020PLOS ONE

Dear Dr. Penn,

Thank you for submitting your manuscript to PLOS ONE. After careful consideration, we feel that it has merit but does not fully meet PLOS ONE’s publication criteria as it currently stands. Therefore, we invite you to submit a revised version of the manuscript that addresses the points raised during the review process.

I read your paper with interest -- this is a quality paper. However, to expedite the publication of this paper, while I’m working on finding knowledgeable reviewers, I’d like to share with you some my thoughts and comments.

1. Interpretation of OA and VB is arbitrary – they are just factors. Can’t we interpret these quantities as factors contributing to playing home versus guest match? The problem with of your interpretation, although plausible, should be mentioned.

2. I believe that there is even a simple model to derive the teams standing I want to share with you using the average of the score difference for each team. Indeed, let rA denote the standing/rank of team A and dAB denote the difference in score A played against B, that is, if the score was 1:3 we have dAB = -2. The major assumption made is that dAB is proportional to the difference in teams standing, dAB = a(rA – rB) + epsAB where a>0 is the scale parameter and epsAB is the error term with zero mean and constant variance. In fact, we may forget about a and simply rewrite the model as dAB = rA – rB + epsAB. Indeed, after rA derived we can make the transformation (rA – min)/(max-min). The above is a linear model. Assuming that ∑ rA = 0 we arrive at the simple solution rA = ∑BdAB /nA where nA is the number of matches played by team A. This model has advantages: (a) the solution is easy to interpret, (b) unlike your model there is no technical issues such as existence and uniqueness (say, that all teams played against each other – then you PAB = const). (c) This model uses all scores while you use only the total number of scores. Disadvantage is that eps does not have a normal distribution. I believe that the comparison of the linear with the Poisson model to predict standings would be of great interest and it would strengthen your paper.

3. Another probabilistic model for ranking the team is suggested in Demidenko E (2020). Advanced Statistics with Applications in R. New York: Wiley.

Some editorial comments:

1. Refer in the introduction to “football/soccer” at least once because many readers are from US.

2. Use “p-value” not “p value.”

3. For presentation purpose I’d derive equation (3).

We look forward to receiving your revised manuscript.

Kind regards,

Eugene Demidenko, Ph.D.

Academic Editor

PLOS ONE
---

## [Author Response · Author response to Decision Letter 0]

4 Feb 2022

Dear Dr Demidenko,

Thank you for your comments on our manuscript. We have written below our responses to each of the points raised.

Sincerely, on behalf of all the authors

Matthew Penn

Main Comments

1. Interpretation of OA and VB is arbitrary – they are just factors. Can’t we interpret these quantities as factors contributing to playing home versus guest match? The problem with of your interpretation, although plausible, should be mentioned.

We have clarified that OA and VB are independent of which team is playing at home or away. We have provided further justification for our interpretation, while also acknowledging the possibility for different interpretations to be made.

2. I believe that there is even a simple model to derive the teams standing I want to share with you using the average of the score difference for each team. Indeed, let rA denote the standing/rank of team A and dAB denote the difference in score A played against B, that is, if the score was 1:3 we have dAB = -2. The major assumption made is that dAB is proportional to the difference in teams standing, dAB = a(rA – rB) + epsAB where a>0 is the scale parameter and epsAB is the error term with zero mean and constant variance. In fact, we may forget about a and simply rewrite the model as dAB = rA – rB + epsAB. Indeed, after rA derived we can make the transformation (rA – min)/(max-min). The above is a linear model. Assuming that ∑ rA = 0 we arrive at the simple solution rA = ∑BdAB /nA where nA is the number of matches played by team A. This model has advantages: (a) the solution is easy to interpret, (b) unlike your model there is no technical issues such as existence and uniqueness (say, that all teams played against each other – then you PAB = const). (c) This model uses all scores while you use only the total number of scores. Disadvantage is that eps does not have a normal distribution. I believe that the comparison of the linear with the Poisson model to predict standings would be of great interest and it would strengthen your paper.

We have added in a subsection to compare our model to the linear model described above. We have shown how to derive rankings from our model, and plotted them against the rankings from the linear model in our new Figure 12. We have also calculated the mean error (in goal difference) between the predictions from each model and the actual results of Euro 2020. We have highlighted evidence that the models are similar, but that the extra information used by the Poisson model helps it to give better predictions.

3. Another probabilistic model for ranking the team is suggested in Demidenko E (2020). Advanced Statistics with Applications in R. New York: Wiley.

We have added this as a reference, referring to it in the introduction.

Editorial Comments

1. Refer in the introduction to “football/soccer” at least once because many readers are from US.

We have added in a reference to soccer in the introduction.

2. Use “p-value” not “p value.”

We have changed this.

3. For presentation purpose I’d derive equation (3).

We have added the derivation of Equation 3 into the main manuscript. We have hence removed it from the supplementary material file S1.

---

## [Decision Letter · Decision Letter 1]

2 Mar 2022

PONE-D-22-01098R1Analysis of a double Poisson model for predicting football results in Euro 2020PLOS ONE

Dear Dr. Penn,

Thank you for submitting your manuscript to PLOS ONE. After careful consideration, we feel that it has merit but does not fully meet PLOS ONE’s publication criteria as it currently stands. Therefore, we invite you to submit a revised version of the manuscript that addresses the points raised during the review process.

Attached are comments from a Reviewer. I think they are constructive and can be easily addressed. Looking forward to seeing a revised version.

We look forward to receiving your revised manuscript.

Kind regards,

Eugene Demidenko, Ph.D.

Academic Editor

PLOS ONE

Journal Requirements:

Reviewer's comment:

1. If the authors have adequately addressed your comments raised in a previous round of review and you feel that this manuscript is now acceptable for publication, you may indicate that here to bypass the “Comments to the Author” section, enter your conflict of interest statement in the “Confidential to Editor” section, and submit your "Accept" recommendation.

Reviewer #1: (No Response)

2. Is the manuscript technically sound, and do the data support the conclusions?

Reviewer #1: Yes

3. Has the statistical analysis been performed appropriately and rigorously? 

Reviewer #1: I Don't Know

4. Have the authors made all data underlying the findings in their manuscript fully available?

Reviewer #1: Yes

5. Is the manuscript presented in an intelligible fashion and written in standard English?

Reviewer #1: Yes

6. Review Comments to the Author

Reviewer #1: Review of:

[Manuscript PONE-D-22-01098R1

Title: Analysis of a double Poisson model for predicting football results in Euro 2020

Authors: Matthew J. Penn, Christl A. Donnelly]

I have read the paper with interest.

The manuscript topic is of interest for “PLOS ONE”.

The authors offer a work that potentially can improve our knowledge about the topic of interest. I suggest that the authors consider a revision of their work along the following suggestions and questions.

Suggestions and Questions:

1- It is recommended to start the abstract with general information about the topic.

2- It is suggested to discuss more about the findings of this study in the abstract.

3- It is recommended to mention about the applications of this study at the end of abstract:

The findings of this study can help for better understanding of …

4- I strongly recommend the authors to add one paragraph discussing the difference between their work and the previously performed studies in literature. In other words, what is the novelty of this work? I offer the authors to revise the abstract and introduction in order to incorporate the novelty of their work. This change motivates the readers of “PLOS ONE” to study this work with interest.

5- It is suggested to add a figure in Section 1, which presents the general sketch of the problem under study.

6- It is recommended to include a paragraph at the end of introduction to present the steps of the work like:

First, the methods are presented. Then,…

7- What are the advantages and disadvantages of this study? I recommend the authors to highlight this topic.

8- What are the limitations of this study? I recommend the authors to highlight this topic.

9- The boxes around the legends in all the plots should be removed.

10- “Where” should be replaced by “where” for definition of parameters or variables after each equation.

11- It is recommended to add minor ticks (or intervals) on horizontal and vertical axis of all the plots.

12- The title for last section should be changed to “Summary and Conclusions”.

13- It is recommended to show the main remarks of this study in terms of bullets in last section (Summary and Conclusions).

14- It is suggested to add a nomenclature (including alphabetic letters, Greek letters, subscripts, and superscripts).

15- It is recommended to keep the main governing equations in the text of manuscript and move the rest of equations to an appendix.

7. PLOS authors have the option to publish the peer review history of their article (what does this mean?). If published, this will include your full peer review and any attached files.

Reviewer #1: No

---

## [Author Response · Author response to Decision Letter 1]

4 Apr 2022

Dear Reviewer

We have received your comments on our manuscript. We have written below our responses to each of the points raised.

Yours faithfully, on behalf of all the authors

Matthew Penn

Main Comments

1- It is recommended to start the abstract with general information about the topic.

We have rewritten the abstract to start with some background information on the double Poisson model.

2- It is suggested to discuss more about the findings of this study in the abstract.

We have added in some of the findings of this study into the abstract.

3- It is recommended to mention about the applications of this study at the end of abstract:

The findings of this study can help for better understanding of …

This has been added to the end of the abstract.

4- I strongly recommend the authors to add one paragraph discussing the difference between their work and the previously performed studies in literature. In other words, what is the novelty of this work? I offer the authors to revise the abstract and introduction in order to incorporate the novelty of their work. This change motivates the readers of “PLOS ONE” to study this work with interest.

The previous draft contained a paragraph starting “The main novel contribution to this work is...” We have added in the word “novel” to the second sentence of this paragraph to emphasise that the whole paragraph is concerned with the novelty of the study. We have also highlighted the novelty in the abstract.

5- It is suggested to add a figure in Section 1, which presents the general sketch of the problem under study.

We have added in this figure.

6- It is recommended to include a paragraph at the end of introduction to present the steps of the work like:

First, the methods are presented. Then,…

This paragraph has been added into the introduction.

7- What are the advantages and disadvantages of this study? I recommend the authors to highlight this topic.

A paragraph discussing the advantages and disadvantages has been added.

8- What are the limitations of this study? I recommend the authors to highlight this topic.

The limitations are highlighted in the new paragraph discussing the advantages and disadvantages.

9- The boxes around the legends in all the plots should be removed.

The boxes around legends have been removed.

10- “Where” should be replaced by “where” for definition of parameters or variables after each equation.

The authors performed a case sensitive search in the main text, and all supplementary material for the word “Where” and did not find any occurrences. However, they are happy to change any of these that have been missed.

11- It is recommended to add minor ticks (or intervals) on horizontal and vertical axis of all the plots.

These have been added to all of the plots.

12- The title for last section should be changed to “Summary and Conclusions”.

The title for the final section has been changed.

13- It is recommended to show the main remarks of this study in terms of bullets in last section (Summary and Conclusions).

The previous conclusion has been moved to a section entitled “Discussion”. The section “Summary and Conclusions” now contains bullets.

14- It is suggested to add a nomenclature (including alphabetic letters, Greek letters, subscripts, and superscripts).

A nomenclature has been added.

15- It is recommended to keep the main governing equations in the text of manuscript and move the rest of equations to an appendix.

The previous review of this manuscript (by the editor, Eugene Demidenko) requested that the equations in the derivation were placed in the main text. The authors therefore query this comment to the editor - the derivation has been currently left in the main text but can be moved if required.

---

## [Editor Report · Decision Letter 2]

7 Apr 2022

PONE-D-22-01098R2Analysis of a double Poisson model for predicting football results in Euro 2020PLOS ONE

Dear Dr. Penn,

Thank you for submitting your manuscript to PLOS ONE. After careful consideration, we feel that it has merit but does not fully meet PLOS ONE’s publication criteria as it currently stands. Therefore, we invite you to submit a revised version of the manuscript that addresses the points raised during the review process. The paper can be accepted after some minor changes that address the comments of the Reviewer.

We look forward to receiving your revised manuscript.

Kind regards,

Eugene Demidenko, Ph.D.

Academic Editor

PLOS ONE
---

## [Author Response · Author response to Decision Letter 2]

11 Apr 2022

Dear Reviewer

We have received your comments on our manuscript. We have written below our responses to each of the points raised.

Yours faithfully, on behalf of all the authors

Matthew Penn

Main Comments

1- It is recommended to start the abstract with general information about the topic.

We have rewritten the abstract to start with some background information on the double Poisson model.

2- It is suggested to discuss more about the findings of this study in the abstract.

We have added in some of the findings of this study into the abstract.

3- It is recommended to mention about the applications of this study at the end of abstract:

The findings of this study can help for better understanding of …

This has been added to the end of the abstract.

4- I strongly recommend the authors to add one paragraph discussing the difference between their work and the previously performed studies in literature. In other words, what is the novelty of this work? I offer the authors to revise the abstract and introduction in order to incorporate the novelty of their work. This change motivates the readers of “PLOS ONE” to study this work with interest.

The previous draft contained a paragraph starting “The main novel contribution to this work is...” We have added in the word “novel” to the second sentence of this paragraph to emphasise that the whole paragraph is concerned with the novelty of the study. We have also highlighted the novelty in the abstract.

5- It is suggested to add a figure in Section 1, which presents the general sketch of the problem under study.

We have added in this figure.

6- It is recommended to include a paragraph at the end of introduction to present the steps of the work like:

First, the methods are presented. Then,…

This paragraph has been added into the introduction.

7- What are the advantages and disadvantages of this study? I recommend the authors to highlight this topic.

A paragraph discussing the advantages and disadvantages has been added.

8- What are the limitations of this study? I recommend the authors to highlight this topic.

The limitations are highlighted in the new paragraph discussing the advantages and disadvantages.

9- The boxes around the legends in all the plots should be removed.

The boxes around legends have been removed.

10- “Where” should be replaced by “where” for definition of parameters or variables after each equation.

The authors performed a case sensitive search in the main text, and all supplementary material for the word “Where” and did not find any occurrences. However, they are happy to change any of these that have been missed.

11- It is recommended to add minor ticks (or intervals) on horizontal and vertical axis of all the plots.

These have been added to all of the plots.

12- The title for last section should be changed to “Summary and Conclusions”.

The title for the final section has been changed.

13- It is recommended to show the main remarks of this study in terms of bullets in last section (Summary and Conclusions).

The previous conclusion has been moved to a section entitled “Discussion”. The section “Summary and Conclusions” now contains bullets.

14- It is suggested to add a nomenclature (including alphabetic letters, Greek letters, subscripts, and superscripts).

A nomenclature has been added.

15- It is recommended to keep the main governing equations in the text of manuscript and move the rest of equations to an appendix.

The previous review of this manuscript (by the editor, Eugene Demidenko) requested that the equations in the derivation were placed in the main text. The authors therefore query this comment to the editor - the derivation has been currently left in the main text but can be moved if required.

---

## [Editor Report · Decision Letter 3]

4 May 2022

Analysis of a double Poisson model for predicting football results in Euro 2020

PONE-D-22-01098R3

Dear Dr. Penn,

We’re pleased to inform you that your manuscript has been judged scientifically suitable for publication and will be formally accepted for publication once it meets all outstanding technical requirements.

Kind regards,

Eugene Demidenko, Ph.D.

Academic Editor

PLOS ONE
---

## [Editor Report · Acceptance letter]

10 May 2022

PONE-D-22-01098R3 

Analysis of a double Poisson model for predicting football results in Euro 2020 

Dear Dr. Penn:

I'm pleased to inform you that your manuscript has been deemed suitable for publication in PLOS ONE. Congratulations! Your manuscript is now with our production department. 

Kind regards, 

on behalf of

Dr. Eugene Demidenko 

Academic Editor

PLOS ONE